# Experimental Study on Hot Spot Stresses of Curved Composite Twin-Girder Bridges

**DOI:** 10.3390/ma15227920

**Published:** 2022-11-09

**Authors:** Rui Zhao, Yongjian Liu, Lei Jiang, Bowen Feng, Yisheng Fu, Chenyu Zhang

**Affiliations:** 1School of Highway, Chang’an University, Xi’an 710064, China; 2Research Center of Highway Large Structure Engineering on Safety of Ministry of Education, Xi’an 710064, China; 3Department of Civil Engineering, Stony Brook University, Stony Brook, NY 11794, USA

**Keywords:** experimental study, hot spot stress, curved composite twin-girder bridge, fatigue detail, fatigue assessment

## Abstract

Curved composite twin-girder bridges are suitable for mountainous areas, due to their advantages of light self-weight, excellent mechanical performance, and fewer construction requirements. It has been found that many composite twin-girder bridges collapsed due to fatigue failure. However, the literature review showed no relevant studies on the fatigue performance of curved composite twin-girder bridges. Because of this, the specimen of 1:2 scale curved composite twin-girder bridge in accordance with the design scheme of Xizhen Bridge in China was designed and tested. Three possible fatigue details were selected: cruciform connections, transverse attachments, and transverse splices named Class I, Class II, and Class III. For the test data of nominal stress (NS), equations were proposed to convert the strain value into the internal force of the fatigue detail position. The stress caused by torsion accounts for 2.8% of the total stress, which is almost negligible. The fatigue evaluation process based on the hot spot stress (HSS) S-N curve method is presented. The HSS method is more conservative than the NS S-N curve method in predicting the fatigue life of complex structures with high-stress concentrations.

## 1. Introduction

Composite steel plate girder bridges have the advantages of light self-weight, accelerated erection, less construction space requirement, and little environmental influence, which is one of the bridge types suitable for prefabricated construction [1,2]. The composite steel plate girder bridge using a twin-girder cross section is expected to be economical compared with the multi-girder bridge due to the lower material, labor, and fabrication cost. The built-up I-section is normally used in the twin-girder cross-section, which is directly welded by three steel plates. In recent five years, composite twin-girder bridges have been widely used in small and medium-span bridges in China, and the general drawing of such a bridge has been proposed [3]. Also, to meet the requirements in the mountainous area in western China, a general drawing of the curved composite twin-girder bridge is proposed.

It has been demonstrated that the high-stress concentration is easily found at the location of the weld toe in the composite twin-girder bridge due to the coupling influences of poor service conditions, heavy loads, and geometrical discontinuities. Thus, the welded connections are prone to fatigue failure, affecting the durability and safety of the composite twin-girder bridges. The statistics in the literature [4] showed that 85% of composite girder bridges in service were critical to fatigue failure. Hence, it is necessary to consider the fatigue behavior of composite twin-girder bridges.

Fisher et al. [5] summarized the fatigue failure cases of composite girder bridges. They found that the crack would initiate at the location of high-stress concentration, which was due to structural discontinuities or weld defects [4,5,6,7,8]. Many researchers have adopted the hot spot stress (HSS) method to evaluate the fatigue life of composite girder bridges [9,10,11,12,13,14,15,16,17,18]. For example, Melaku et al. [19] used the HSS method and conducted only using finite element method (FEM) models to evaluate the connection of vertical stiffener and web plate. Karabulut et al. [20] measured the HSS in welded cruciform duplex stainless steel joints and presented the HSS S-N curve to improve Eurocode further. Eurocode [21] provides HSS S-N curves for steel plate girder bridge connections in the appendix. However, the test only took out the component of steel plate girder bridges for loading, and the stress concentration of the real bridge may be different from that of the component test due to the size effect. There is no test of HSS of composite twin-girder bridges in the current research, which seriously hinders the popularization and application of composite steel plate girder bridges.

To fill this gap, the specimen of a 1:2 scale curved composite twin-girder bridge in accordance with the design scheme of Xizhen Bridge in China is designed and tested. Three possible fatigue details are selected as follows: cruciform connections, transverse attachments, and transverse splices, which were named as Class I, Class II, and Class III. The HSS and NS for three fatigue details were measured to evaluate the stress concentration. Meanwhile, the Equations of transforming strain into internal force were given. And the axial force, bending moment, and shear force of the cross beam and steel I-girder during the loading process are deduced. The fatigue evaluation process of steel plate composite girder bridge based on the HSS S-N method was presented to predict fatigue life. The fatigue life evaluation results based on the Eurocode using the HSS and NS S-N method were given and compared. Finally, the finite elements (FE) were developed and verified to study the fatigue life of a curved composite twin-girder bridge.

## 2. Overview of the Test

### 2.1. Specimen Design

As shown in Figure 1, a 1:2 scale continuous curved composite twin-girder bridge with a span arrangement of 2 × 17.5 m was tested. It was composed of two built-up I-girders and a concrete deck slab through shear studs, with a curve radius of 200 m.

Figure 2 presents the cross-section view of the specimen. The total height of the superstructure was 1100 mm, with a 900 mm height of steel I-girder and 200 mm thickness of concrete deck slab. The space between two I-girders was 3350 mm, and the width of the concrete deck slab was 6375 mm. During construction, the concrete deck slab was made of precast segments with a length of 1400 mm full width, and 350 mm wet joints of cast-in-place concrete were used between the precast segments. Figure 3 shows the rebar arrangement of full-width wet joints.

Table 1 lists the dimensions of the steel I-girder and cross beam. Variable cross-sections of steel I-girders were adopted. For the cross-section in the region of 750 mm close to the middle support, the widths of the upper and lower flanges were 500 mm and 580 mm, respectively. For the cross-section between 750 mm and 2500 mm away from the middle support, the widths of the upper and lower flanges varied from 500 mm and 580 mm to 400 mm and 480 mm, respectively. Accordingly, for the cross-section in the region of 2500 mm away from the middle support, the widths of the upper and lower flanges were 400 mm and 480 mm, respectively. Also, variable plate thicknesses of steel I-girders along the longitudinal direction of the bridge were adopted to make full use of the material (Figure 2a). The thicknesses of the top flanges were 11 mm, 14 mm, 16 mm, and 24 mm for sections A, B, C, and D, respectively. The thicknesses of the bottom flanges were 20 mm, 27 mm, 19 mm, and 30 mm for sections A, B, C, and D, respectively. The thicknesses of the webs were 10 mm, 10 mm, 12 mm, and 14 mm for sections A, B, C, and D, respectively. Besides, two cross beams with the same dimensions were arranged in each span of the bridge at the middle and end supports, namely middle and end cross beams, respectively, while four cross beams were arranged along the longitudinal direction with a distance of 3500 mm for each span, namely small cross beams. The detailed dimensions of cross beams can be found in Table 1, and the corresponding detailed dimensions of concrete deck slabs can be found in Figure 2. The arrangement of rebars in the concrete deck slab is shown in Figure 2e, where the diameter of rebars was 14 mm and 8 mm for longitudinal rebars and stirrups. The spacing of the stirrups was 62.5 mm.

### 2.2. Possible Fatigue Details

In this test, three possible fatigue details named Class I, Class II, and Class III were selected, as shown in Figure 4. The hot spot stress distributions for these three fatigue details were measured to evaluate the degree of stress concentration and then predict the fatigue life of the curved composite twin-girder bridges based on the current specification. The descriptions of three fatigue details are presented as follows:Class I: cruciform connections. Cross beams are connected to the web of steel I-girder using cruciform connections. Toe failure may occur in full penetration butt welds;Class II: transverse attachments. Vertical stiffeners are welded to the web of steel I-girder with a cope hole. Toe failure in full penetration butt welds may occur in the web or bottom flange of steel I-girders;Class III: transverse splices. Two segments with variable thicknesses are connected using butt welds. Toe failure in full penetration butt welds or partial penetration butt welds may occur in the web or flanges of steel I-girders.

### 2.3. Material Properties

As specified in Chinese Standard JTG D64-2015 [22], the weathering steel of grade Q345qDNH was adopted for steel I-girders and cross beams. According to Chinese Standard GB/T 228-2010 [23], a coupon test was conducted, and the measured steel properties are summarized in Table 2. Also, as specified in Chinese Standard JTG 3360-2018 [24], the concrete of grade C50 and C55 were adopted for the concrete deck slab and the wet joint between deck segments, respectively. As recommended by Chinese Standard GB/T 50081-2002 [25], 150 mm× 150 mm× 150 mm concrete cube specimens were tested, and the measured concrete properties are presented in Table 3.

### 2.4. Loading Scheme

Figure 5 shows the loading scheme and test setup, which consists of the reaction frame, jack actuators, and ground anchors. The two-span continuous beam was tested under ten points bending by applying the loading to the middle locations of each span, as shown in Figure 5a. For every loading position, two jack actuators were used to apply symmetrical compressive loading to the cross-section, as shown in Figure 5b. During the loading process, the values of compressive forces were monitored by pressure sensors to ensure synchronous and uniform loading applied to four points.

Before the test, preloading was applied to ensure complete contact between the specimen and the loading device to eliminate the test errors. After the preloading scheme, the loading was applied step by step using the strategy to control the value of the loading. It is assumed that the specimen is in the elastic stage when the value of applied loading is below 65% of the predicted ultimate loading; thus, loading with the increment of 40 kN was applied to the specimen. On the contrary, loading with an increment of 20 kN was applied to the specimen for the plastic stage. The test was halted when the value in the pressure sensor could not increase, which was considered the ultimate strength of the bridge.

### 2.5. Measurement Arrangements

In the test, both hot spot stresses and nominal stresses in fatigue details classes I to III were measured. Considering the curvature radius existing in this bridge, larger loading effects would be found in the external steel I-girder [26], so the hot spot stress strains and nominal stress strain measurements were arranged in it.

The location of fatigue details was selected according to the loading scheme of obvious stress. Figure 4 shows the location arrangements of fatigue details in classes I to III at a girder length. Figure 6 shows the measurement arrangements of hot spot stresses in fatigue detail in classes I to III. A total of 17 strip strain gauges were arranged at locations A to Q, where crack initiations could occur [27]. For fatigue detail Class I, strip strain gauges were installed at locations A to F to measure hot spot stresses at the weld toe between the small cross beam and the web of steel I-girder. For fatigue detail, Class II strip strain gauges were installed at locations G to I to measure hot spot stresses at the weld toe between the vertical stiffener and steel I-girder. For fatigue detail, Class III strip strain gauges were installed at locations J to Q to measure hot spot stresses at the weld toe between two segments of steel I-girder. As recommended by the literature [27], hot spot stress should be evaluated through the quadratic extrapolation method (Figure 7). The strip strain gauge consisted of three integrated strain gauges spaced 4 mm apart arranged in the hot spot region, where the first gauge should be placed 4 mm away from the weld toe as specified in IIW-XV-E [28].

Figure 8 shows measurement arrangements for nominal stress in fatigue details classes I to III. For each fatigue detail, three columns of nominal strain gauges with a space of 100 mm were installed (Figure 8a–c). In this way, the bending moment at the weld toe can be evaluated by linear extrapolation according to the linear elasticity theory. Besides, four nominal strain gauges were installed on the lower flange of the steel I-girder at mid-span to evaluate the torque (Figure 8d).

## 3. Test Result and Analysis

### 3.1. Load-Displacement Curve

Figure 9 presents the load-displacement curve measured at the loading point of the external arc steel I-girder. Table 4 shows loads corresponding to each feature point in the loading process. The test results show that the first feature point was found at 300 kN, the initial crack appearing in the concrete deck slab. Then, the crack in the concrete deck slab propagated to the full width with a length of 280 cm at the applied loading of 1350 kN. At this time, there is only one crack, and the position of the crack is at the center support, which has little influence on the mid-span load-displacement curve. Also, the yield of the lower flange of the external arc steel I-girder and the buckling of the whole external arc steel I-girder were found at the applied loading of 1811 kN and 2603 kN, respectively. As shown in Figure 9, the slope of the load-displacement curve basically remains unchanged up to the loading of 1811 kN, which means the test specimen was in the elastic range. The buckling in this manuscript refers to the buckling of the steel girder, and the buckling state is shown in Figure 10. Finally, the peak applied loading of 2627 kN was measured. As shown in Figure 9, the slope of the load-displacement curve basically remains unchanged up to the loading of 1811 kN, which means the test specimen was in the elastic range.

### 3.2. Nominal Stresses

#### 3.2.1. Stress Analysis

Strain (*ε*) can be converted to stress (*σ*) by Hooke’s law [14] as follows:(1)σ=Eε

The curved bridge has torque in internal force compared with the straight bridge, so this paper first discusses torsion. Torsion is divided into free torsion and constrained torsion according to whether there is a constraint. Free torsion does not produce normal stress; confined torsion produces normal stress. The normal stress of a steel girder is produced by bending moment, axial force, and constraint torsion. Figure 11a,b shows the stress distribution caused by torsion, bending moment, and axial force of the double main beams. The normal stress generated by torsion is antisymmetric centered on the axis of symmetry. According to the research of Shihua Bao [29], the normal stress caused by torsion can be estimated according to the four measuring points of the lower flange of the curved composite twin-girder bridge. Figure 11c shows the distribution of normal stress generated by torsion, bending moment, and axial force when P = 1000 kN. It can be seen that the normal stress caused by torsion accounts for 2.8% of the total stress, which is almost negligible. Therefore, the influence of torsion is not considered in this paper.

The force analysis diagram of a curved composite twin-girder bridge is shown in Figure 11. The stress distribution can be analyzed by Equations (2)–(6), and the internal force distribution of the steel I-girder and the cross beam can be obtained. The shear stress on the cross-section is not measured and is derived using the *y*-axis force balance and bending moment balance equations. The relevant equations are as follows:(2)$$\sum F_{x}=N_{R}-N_{L}=0$$
(3)$$\sum F_{y}=\int_{-h_{0}}^{h-h_{0}}\tau_{L}d\tau -\int_{-h_{0}}^{h-h_{0}}\tau_{R}d\tau =V_{L}-V_{R}=0$$
(4)$$\sum M=M_{L}+\int_{-h_{0}}^{h-h_{0}}\tau_{L}d\tau \cdot x_{1}-M_{R}=0$$
(5)$$\sigma_{s}=\frac{M_{i}y}{I_{xj}}+\frac{N_{i}}{A_{j}}(i=L,R;j=0,1)$$
(6)$$\sigma_{c}=\frac{1}{\alpha_{E}}(\frac{M_{i}y}{I_{xj}}+\frac{N_{i}}{A_{j}})(i=L,R;j=0,1)$$
where the meaning of each symbol is shown in Figure 12, *σ*_s_ is the stress on the steel I-girder, *σ*_c_ is the stress on the concrete slab. *I*_xj_ and *A*_j_ are the equivalent section moments of inertia and area, respectively. *j* = 0 represents the steel plate composite girders in the positive bending moment area. *j* = 1 represents the steel plate composite girders in the negative bending moment area and a small cross beam. In addition, the force analysis assumes that the tension is positive and the compression is negative. Make the bending moment in tension at the lower part of the girder positive, and make the bending moment of tension at the upper part of the girder negative. The clockwise shear is positive, and the counterclockwise shear is negative.

In solving the internal force, the composite section composed of steel and concrete should be equivalent to the same material. To simplify the calculation, the concrete bridge slab is equivalent to steel, and the equivalence principle is that the height and strain of the concrete slab before and after the equivalent are unchanged. The equivalent steel cross-sectional area can be further expressed as:(7)$$A_{0}=b_{b}t_{b}+h_{w}t_{w}+b_{t}t_{t}+b_{e}h_{c1}/\alpha_{E}$$
where the meaning of each symbol is shown in Figure 13. *α*_E_ is the ratio of the elastic modulus of steel to concrete (*α*_E_ = *E*_s_/*E*_c_).

The calculation of the distance from the neutral axis to the bottom of the composite twin-girder bridge in a positive bending moment area (*h*_00_) and the inertia moment (*I*_x0_) is as follows:(8)$$h_{00}=\frac{1}{A_{0}}\left[b_{b}t_{b}^{2}/2+h_{w}t_{w}\left(t_{w}/2+t_{b}\right)+b_{t}t_{t}\left(t_{t}/2+h_{w}+t_{b}\right)+b_{e}h_{c1}(h-h_{c1}/2)/\alpha_{E}\right]$$
(9)$$I_{x0}=\frac{1}{12}\left(b_{b}t_{b}^{3}+t_{w}h_{w}^{3}+b_{t}t_{t}^{3}\right)+b_{b}t_{b}{\left(h_{00}-\frac{t_{b}}{2}\right)}^{2}+h_{w}t_{w}{\left(h_{00}-t_{b}-\frac{h_{w}}{2}\right)}^{2}+b_{t}t_{t}{\left(h_{w}+\frac{t_{t}}{2}+t_{b}-h_{00}\right)}^{2}+I_{c}$$
(10)$$I_{c}=\frac{1}{12\alpha_{E}}b_{e}h_{c1}^{3}+\frac{b_{e}h_{c1}}{\alpha_{E}}{\left(h-h_{00}-h_{c1}/2\right)}^{2}$$

The calculation of the distance from the neutral axis to the bottom of the composite twin-girder bridge in the negative bending moment area and small cross beams (*h*_01_), the inertia moment (*I*_x1_), and cross-sectional area (*A*_1_) is as follows:(11)$$A_{1}=b_{b}t_{b}+h_{w}t_{w}+b_{t}t_{t}$$
(12)$$h_{01}=\frac{1}{A_{0}}\left[b_{b}t_{b}^{2}/2+h_{w}t_{w}\left(t_{w}/2+t_{b}\right)+b_{t}t_{t}\left(t_{t}/2+h_{w}+t_{b}\right)\right]$$
(13)$$I_{x1}=\frac{1}{12}\left(b_{b}t_{b}^{3}+t_{w}h_{w}^{3}+b_{t}t_{t}^{3}\right)+b_{b}t_{b}{\left(h_{01}-\frac{t_{b}}{2}\right)}^{2}+h_{w}t_{w}{\left(h_{01}-t_{b}-\frac{h_{w}}{2}\right)}^{2}+b_{t}t_{t}{\left(h_{w}+\frac{t_{t}}{2}+t_{b}-h_{01}\right)}^{2}$$

The bending moment is linearly distributed along the axial direction of the composite twin-girder bridge and small cross beam. The bending moment in weld toe can be obtained correspondingly through the linear extrapolation value of the bending moment derived from the strains of the three rows of nominal strains.

#### 3.2.2. Internal-Force Analysis

When the crack distribution area of the concrete slab gradually expands, the lower flange plate of the outer arc steel girder will gradually yield, and the stress on the section redistributes. Combined with the change of structural stiffness in Figure 9, it can be concluded that the curved composite twin-girder bridge must be at the elastic stage when the load is in the range of 0–1800 kN. Therefore, this experiment records nominal strains and hot spot strain distribution of classes I–III from loading to 1800 kN.

Figure 14 The strain of potential fatigue details in curved composite twin-girder bridge and small cross beam. Figure 15 shows the curves of the bending moment, shear force, and axial force varying with the load in classes I–III. As shown in Figure 14, the left side of fatigue detail I is A1, and the right side of fatigue detail I is A3. The left side of fatigue detail II is D1, and the right side of fatigue detail II is D3. The left side of fatigue detail III is G1, and the right side of fatigue detail III is G3. Figure 15a shows the bending moment of Class I; the value of the left bending moment is 20.7% larger than the right bending moment. To keep the bending moment balance, the difference in bending moment is compensated by a couple of shear stresses on both sides of Class I. The average value of the difference between the axial forces on the left and right sides of the small beam is only 7%, so the difference can be ignored. It is considered that the axial force satisfies the equilibrium equation. Figure 15d–f shows the internal force distribution of Class II and Class III. The difference between the bending moments at the left and right ends of Class II can reach 85%. This results in a large shear force on both sides of Class II to compensate for the large bending moment difference. When P = 1000 kN, the shear force on both sides of Class II reaches 648.1 kN. The difference between the bending moments at the left and right ends of Class III is only 18.7%. Therefore, the shear force caused by compensating for the difference in bending moment is also smaller. When P = 1000 kN, the shear force on both sides of Class III is only 795.5 kN. The section strain conforms to the plane-section assumption. Both steel and concrete are considered ideal elastic materials.

### 3.3. Hot Spot Stresses

When the load reaches 1000 kN, the steel plate composite girder bridge is in the elastic stage. Figure 15 shows the internal force of the steel plate composite girder, and the small cross beam increases linearly with the increase of the load, indicating the strain test is stable. Figure 16 shows the internal force distribution of classes I–III under *P* = 1000 kN. Consistent with the analysis results in Section 3.2.1, the internal forces on both sides of each isolator satisfy the balance equation of force and bending moment. Meanwhile, the hot spot strain data of classes I–III measured at *P* = 1000 kN are analyzed. The hot spot strain and stress results are separately listed in Table 5 and Table 6.

Three gauges are attached to the plate edge at reference points 4, 8, and 12 mm distant from the weld toe. The hot spot strain is determined by quadratic extrapolation as follows:(14)$$\sigma_{hs}=3\sigma_{4\mathrm{mm}}-3\sigma_{8\mathrm{mm}}+\sigma_{12\mathrm{mm}}$$

Combined with the force analysis in Figure 15, the hot spot strain gauges arranged in Figure 6 satisfied the analysis requirements. For Class I, the maximum HSS in flange and web occurs at B and E, which are 121.2 MPa and 157.1 MPa, respectively. In addition, the value of HSS on the inner side of the flange of the small beam is larger than that on the outer side. The stress of the complex position of the weld is more likely to cause stress concentration. As shown in Figure 17, the HSS of classes I–III is not only related to the geometrical structural details but also related to the nominal stress level. For Class II, since the fatigue detail is located at the mid-span loading point, the larger NS causes the HSS level near the weld to be higher than that of Class I and Class III. After welding the stiffener on the steel girder, the maximum HSS occurs at point G, and the HSS value is as high as 425.0 MPa. For Class III, the maximum HSS of the steel girder flange and web occurs at points L and N, which are 90.0 MPa and 125.1 MPa, respectively. The HSS level of points P and Q is also higher.

## 4. Fatigue Assessment Method Based on Hot Spot Stress

In bridge engineering, the potential fatigue details of curved composite twin-girder bridges are usually high cycle fatigue, whose stress is mainly linear elastic. Therefore, the fatigue assessment process of a curved composite twin-girder bridge based on the HSS S-N curve is proposed, which can be divided as follows:*(1)* *The Fatigue Critical Member is Determined.*

The fatigue critical member is steel tension members or components whose failure would be expected to result in a partial or full collapse of the bridge [30]. Collect the fatigue detection cases of the same bridge type and summarize the details of fatigue prone to occur. Analyze the actual stress state of the bridge in the operation stage, find the components often under tensile stress, pay attention to the details of fatigue vulnerability, and determine the fatigue critical components of the bridge.

*(2)* 
*Determine the HSS S-N Curve Corresponding to the Fatigue Critical Member.*


There are three methods to determine the HSS S-N curve. ① Obtained the HSS S-N curve through a full-scale structural detail test and statistical processing. ② Directly obtained the HSS S-N curve with similar structural details in the relevant specifications. ③ Deduced the HSS S-N curve via stress concentration coefficient and existing NS S-N curve of similar structural details.

*(3)* 
*Obtain the Fatigue Stress Spectrum and Stress Cycle of the Fatigue Critical Member.*


The fatigue stress spectrum can be obtained from the following methods. ① Field measurement method. The stress history of a period is measured on the real bridge, and then the stress spectrum is obtained. ② Method of simulation. The fatigue evaluation load model of the bridge was obtained from traffic observation, archival data investigation, and statistical prediction of field-measured data. Then the stress spectrum of the real bridge is obtained by simulating the stress history. The rain flow counting method transforms the stress course into a series of cyclic stress cycle processes, reflects certain mechanical principles in the counting principle, and is widely used in engineering [31].

*(4)* 
*The Fatigue Life of the Fatigue Critical Member is Calculated by fatigue damage cumulative theory.*


According to Eurocode 3 [21], the fatigue life calculation formula is determined. Equation (15) can be used to calculate the fatigue life when Δ*σ*_i_ ≥ Δ*σ*_D_.
(15)$$N_{i}=5\times 10^{6}{\left(\frac{\Delta \sigma_{D}}{\Delta \sigma_{i}}\right)}^{3}$$

Equation (16) can be used to calculate the fatigue life when Δ*σ*_L_ < Δ*σ*_i_ < Δ*σ*_D_.
(16)$$N_{i}=5\times 10^{6}{\left(\frac{\Delta \sigma_{D}}{\Delta \sigma_{h}}\right)}^{5}$$
where Δ*σ*_i_ (i = 1, 2, ……, z) is the HSS amplitude of the fatigue critical member. *N*_i_ is the total number of cycles corresponding to the fatigue fracture of the fatigue critical member corresponding to Δ*σ*_i_. Δ*σ*_D_ is the constant amplitude fatigue limit. Δ*σ*_L_ is the cut-off of the fatigue curve. When Δ*σ*_i_ ≤ Δσ_L_, the fatigue critical member will not be damaged.

## 5. Comparison between HSS and NS Methods for Fatigue Assessment

To find out the difference between the HSS method and the NS method, the fatigue life of the curved composite twin-girder bridge is evaluated according to the S-N curves of HSS and NS in Eurocode [21]. Table 7 shows classes I–III corresponding to the category with fatigue details in Eurocode.

The fatigue life of classes I–III can be predicted according to the fatigue assessment process and the test results in Section III. Table 8 shows the number of cycles predicted by the NS method and the HSS method for classes I–III. The ratio of the HSS amplitude of NS amplitude, i.e., the stress concentration factor (SCF), represents the degree of stress concentration. According to CIDECT [32], SCF can be calculated as follows:(17)$$SCF=1.2\frac{\sigma_{h}}{\sigma_{n}}$$

The SCF for Class I was 4.68, but the number of cycles predicted by the NS method (*N*_n_) was 67.3 times that predicted by the HSS method (*N*_h_). The SCF of Class II is 3, *N*_n_ is 8 times that of *N*_h_. The SCF of Class III is 2.88, *N*_n_ is 3.5 times that of *N*_h_. It can be found that SCF is a number greater than 1, and *N*_h_ has always been smaller than *N*_n_. In addition, the larger the SCF is, the larger the difference between *N*_n_ and *N*_h_ is. This indicates that the HSS method is more conservative in predicting the fatigue life of complex structures with high-stress concentrations.

## 6. Bridge Evaluation

The ABAQUS software was used to conduct a 3-D FE model of a 1:2 scale bridge. By comparing the FE model with the test, a suitable modeling method for simulating the curved composite steel plate twin-girder bridge was determined. The FE model of the real bridge was also carried out, and the fatigue performance of the real bridge was evaluated using the HSS evaluation process.

The FE model is shown in Figure 18. A 4-node doubly curved thin shell (S4R) element was used for steel girders and beams, and an 8-node linear brick (C3D8R) element was used for concrete deck slabs and studs. For the contact relationship between the steel twin-girder and concrete deck slab, a friction coefficient of 0.3 was set in the tangential direction, and hard contact was set in a normal direction. The loading was consistent with the test loading. The shear studs between the steel girder and the concrete deck were simulated with solid elements. The shear studs were tied to the steel beam and embedded into the concrete deck.

The modeling process completely simulated the test, and the simulated test loading was carried out. To analyze the hot spot stress in fatigue details I–III, the mesh near the weld of fatigue details used local grid refinement, and a 2 mm mesh was used for calculation. To save computational efficiency, a coarse mesh with a mesh size of 10 mm was used for parts far from fatigue details. To make the finite element and the test correspond one to one, the hot spot stress was also solved by quadratic extrapolation using the same method as in Section 2.5. This paper used a load of 1000 kN for verification. Since there was only one small crack on the test bridge at this time, and it had little influence on the load-displacement curve, this paper ignored the existence of a crack in the modeling process.

Figure 19a compares the FE and the test value of the load-displacement curve. The curves of the test values and FE fitting values are in good agreement, and the maximum difference between them is only 5%. Figure 19b shows the comparison between the FE and test values of the HSS of classes I–III, the mean value *μ* = 0.975, mean square deviation *σ* = 0.024, coefficient of variation *σ*/*μ* = 0.024, and the maximum difference between them is 10.2%. In conclusion, the FE model has high computational accuracy.

After the accuracy of the FE model was verified, the fatigue vehicle model IV given by Eurocode was used to analyze fatigue performance for the real bridge. The fatigue vehicle model shown in Table 9 consists of five kinds of lorry, and the proportion of each truckload was determined according to the road grade. The Dload subroutine was written in Fortran to apply the load to the FE model, and the HSS amplitudes at classes I–III were obtained. The maximum HSS amplitudes of the bridge at classes I–III are shown in Table 10. It can be seen from the figures that the HSS amplitude of Class II was significantly higher than that of classes I and III, indicating that Class II is prone to fatigue cracking. With the change of fatigue car, classes I, II, and III showed the same fluctuation trend, and the HSS amplitude increased with the increase of equivalent axial loads. Meanwhile, the HSS amplitude generated by the five-axis fatigue vehicle is the largest. The remaining fatigue life of the bridge was evaluated by the maximum HSS amplitude of each detail, and it was found that the life of each detail of the bridge was more than 2 million times.

## 7. Conclusions

In this paper, the specimen of a 1:2 scale curved composite twin-girder bridge in accordance with the design scheme of Xizhen Bridge in China was designed and tested. According to the test loading arrangement, HSS and NS measuring points were arranged in the tension position of the most unfavorable position of each detail. The Equations of transforming strain into internal force were given, and the axial force, bending moment, and shear force of the cross beam and steel I-girder during the loading process were deduced. The fatigue evaluation process of steel plate composite girder bridge based on the HSS S-N method was presented to predict fatigue life. Finally, the fatigue life evaluation results using the HSS and the NS S-N method were given based on the Eurocode.

(1). Three possible fatigue details were selected as potential bridge fatigue-critical details. Cruciform connections, transverse attachments, and transverse splices were named as Class I, Class II, and Class III.

(2). For the test data of NS, Equations (2)–(4) were proposed to convert the strain value into the internal force for classes I–III. For Class I, the maximum HSS in flange and web occurred at B and E, which were 121.2 MPa and 157.1 MPa, respectively. The value of HSS on the inner side of the flange of the small beam was larger than that on the outer side. The stress of the complex position of the weld was more likely to cause stress concentration. For Class II, since the fatigue detail is located at the mid-span loading point, the larger NS caused the HSS level near the weld to be higher than that of classes I and III. After welding the stiffener on the steel girder, the maximum HSS occurred at point G, and the HSS value was as high as 425.0 MPa. For Class III, the maximum HSS of the steel girder flange and web occurred at points L and N, 90.0 MPa and 125.1 MPa, respectively.

(3). The stress caused by torsion accounts for 2.8% of the total stress, which is almost negligible. Therefore, the influence of torsion is not considered in this paper.

(4). The fatigue evaluation process based on the HSS S-N curve method was presented, and some parameters, such as variable amplitude fatigue and load action time, were introduced to make the evaluation process suitable for real bridge evaluation.

(5). The SCF for Class I was 3.9, but the number of cycles predicted by the NS method (*N*_n_) was 92.3 times that predicted by the HSS method (*N*_h_). The SCF of Class II is 2.5, *N*_n_ is 8 times that of *N*_h_. The SCF of Class III is 1.7, *N*_n_ is 3.5 times that of *N*_h_. It can be seen that SCF is a number greater than 1, and *N*_h_ is always smaller than *N*_n_. In addition, the larger the SCF is, the larger the difference between *N*_n_ and *N*_h_. This indicates that the HSS method is more conservative in predicting the fatigue life of complex structures with high-stress concentrations.

(6). The HSS amplitude of Class II was significantly higher than that of classes I and III, indicating that Class II is prone to fatigue cracking. The remaining fatigue life of the actual bridge was evaluated by the maximum HSS amplitude of each potential fatigue detail, and it was found that the remaining life of the bridge is more than 2 million times.

## Figures and Tables

**Figure 1 materials-15-07920-f001:**
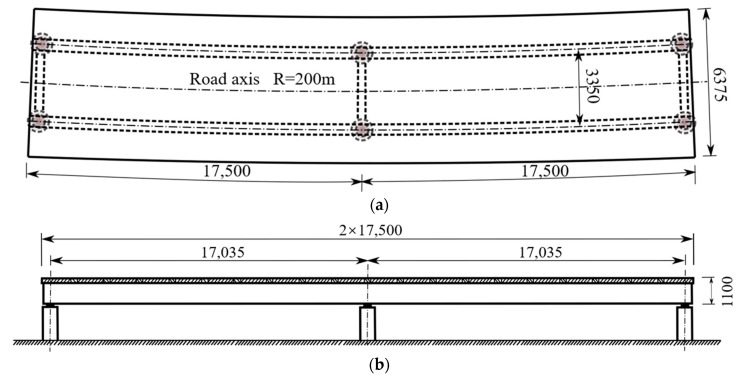
General arrangement of the bridge. (**a**) Plan view; (**b**) Longitudinal view. (Unit: mm).

**Figure 2 materials-15-07920-f002:**
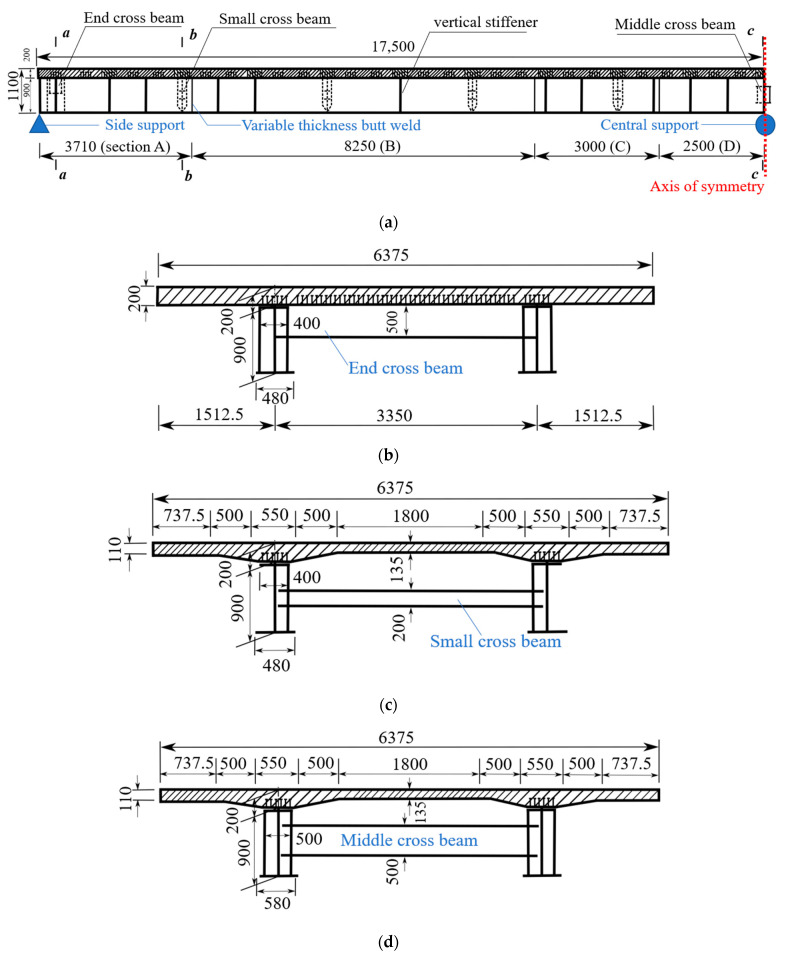
Cross-section view of the bridge (Unit: mm). (**a**) Longitudinal view; (**b**) Section *a*-*a*; (**c**) Section *b*-*b*; (**d**) Section *c*-*c*; (**e**) Rebar arrangement.

**Figure 3 materials-15-07920-f003:**
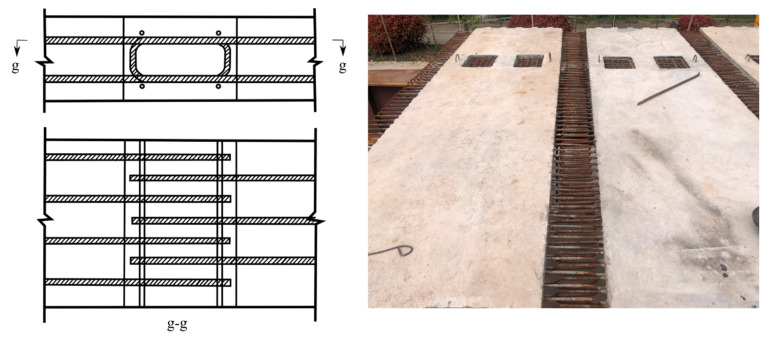
Rebar arrangement of full-width wet joints.

**Figure 4 materials-15-07920-f004:**
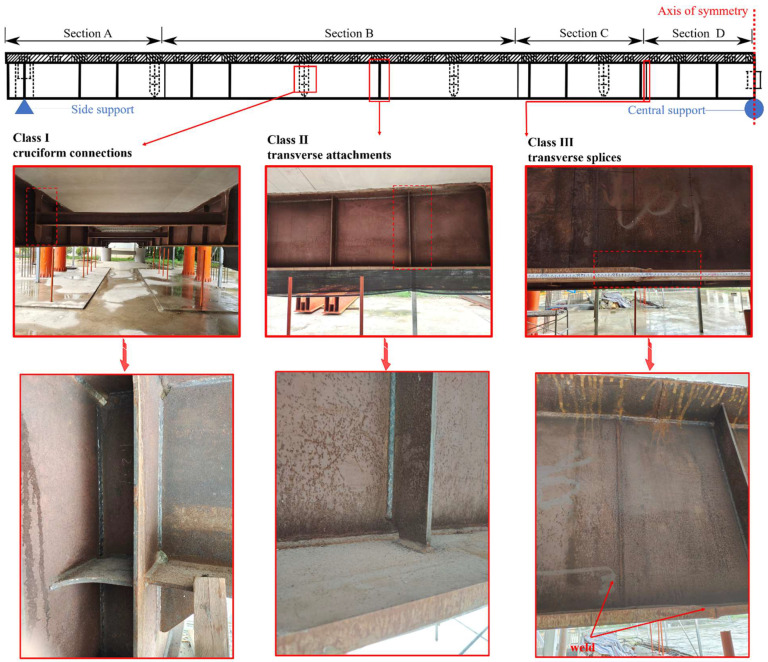
Potential fatigue details.

**Figure 5 materials-15-07920-f005:**
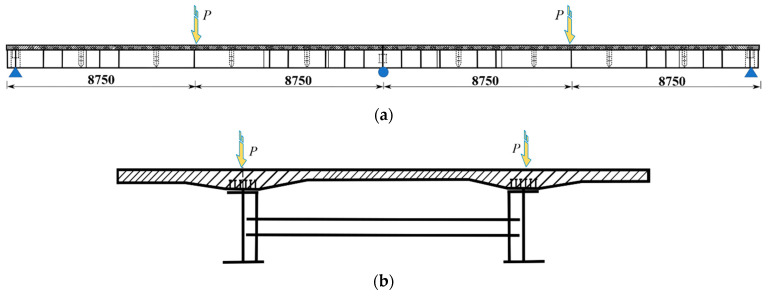
Test setup. (**a**) Elevation view of loading scheme; (**b**) Cross-section view of loading scheme; (**c**) Photo of test setup.

**Figure 6 materials-15-07920-f006:**
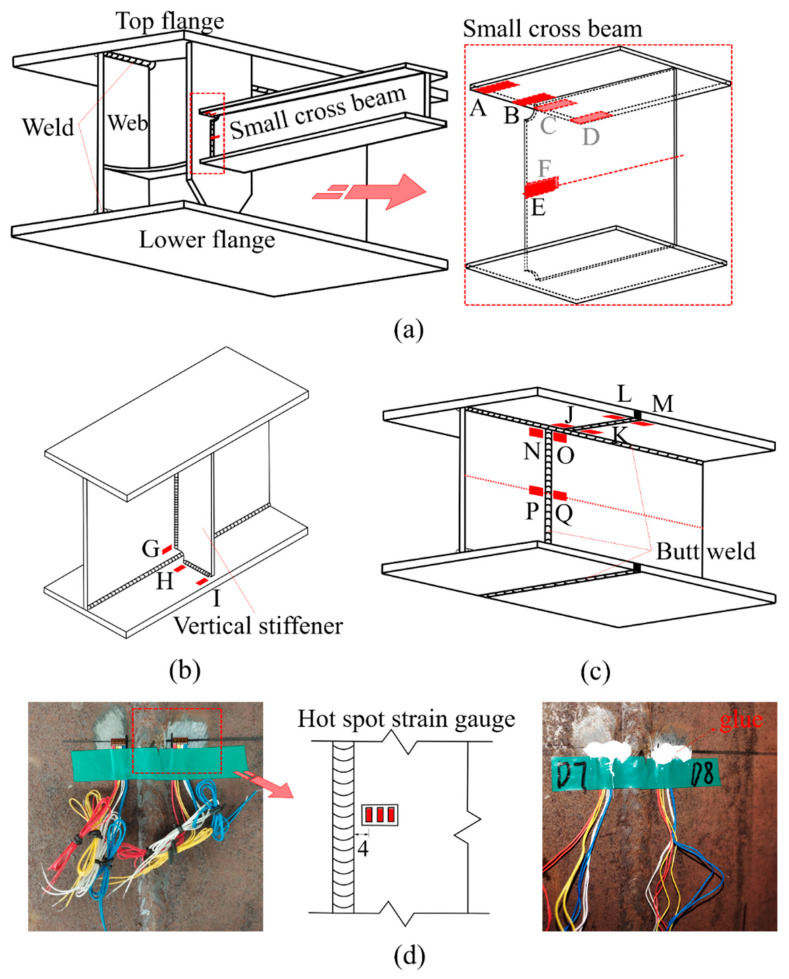
Measurement arrangements for hot spot stress in fatigue details classes I to III.

**Figure 7 materials-15-07920-f007:**
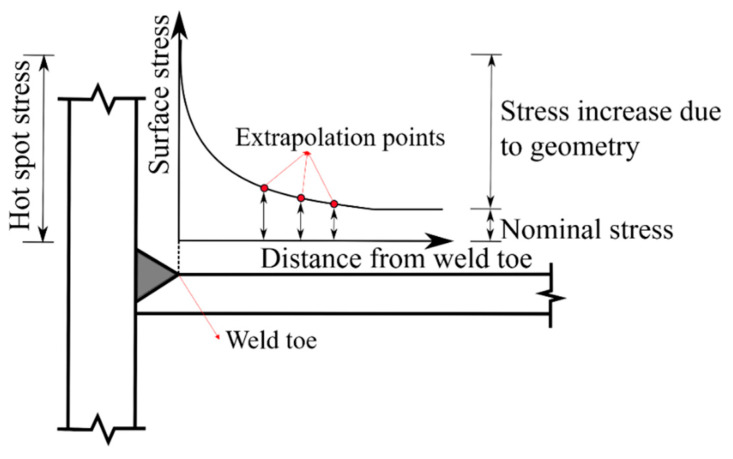
Quadratic extrapolation method.

**Figure 8 materials-15-07920-f008:**
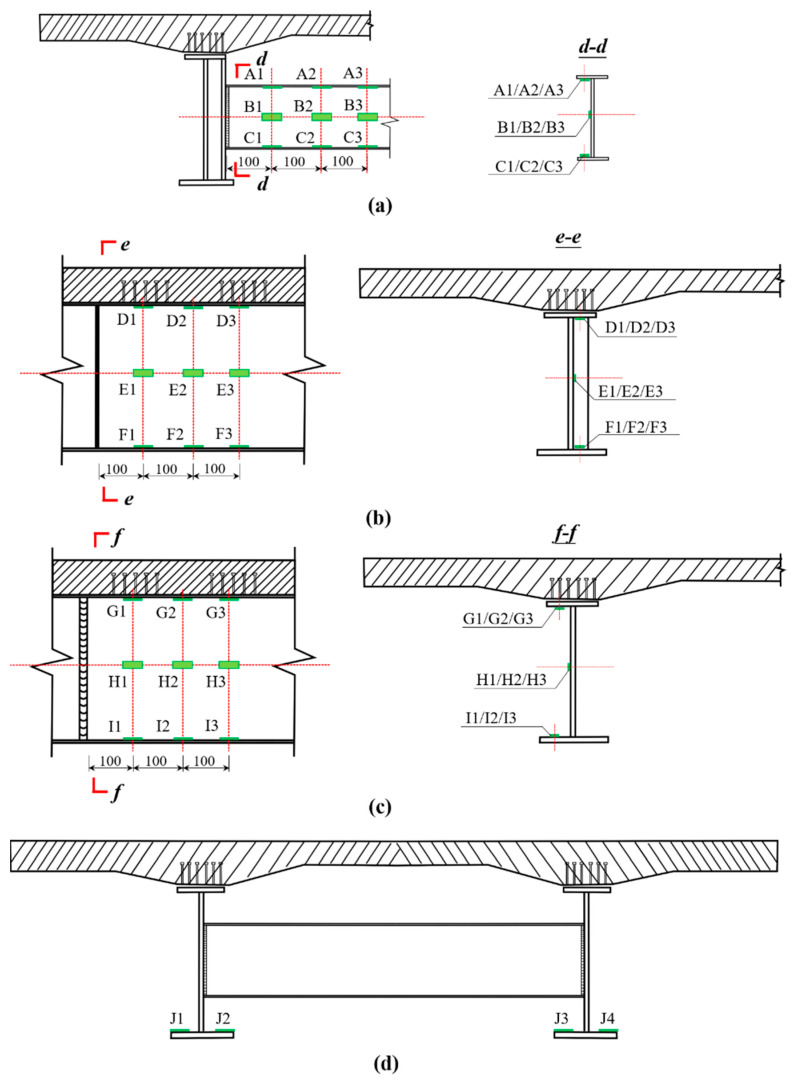
Measurement arrangements for nominal stresses in fatigue details classes I to III. (**a**) Class I; (**b**) Class II; (**c**) Class III; (**d**) lower flange of steel I-girder at mid-span.

**Figure 9 materials-15-07920-f009:**
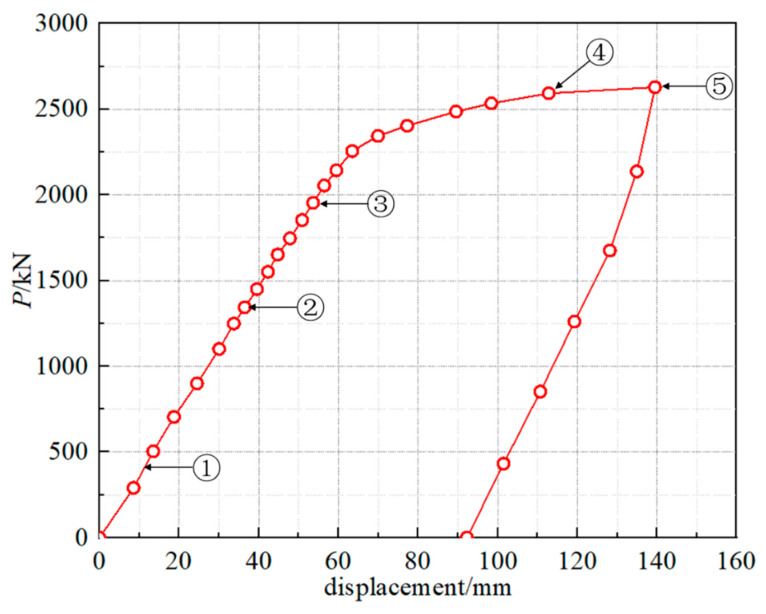
Load-displacement curve measured at the loading point of the external arc steel I-girder.

**Figure 10 materials-15-07920-f010:**
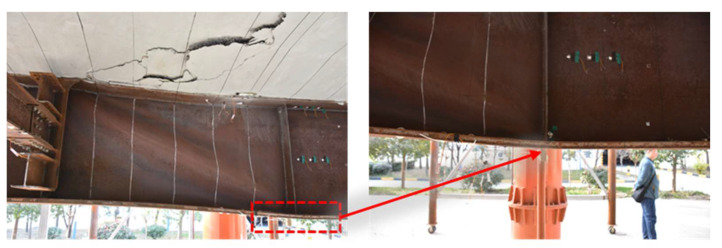
Buckling of the steel girder.

**Figure 11 materials-15-07920-f011:**
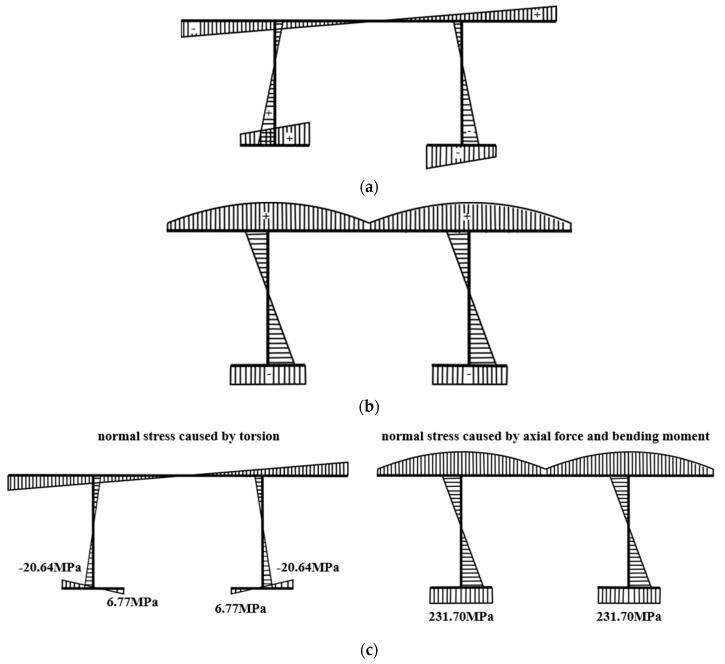
Normal stress distribution diagram of curved composite twin-girder bridge. (**a**) Normal stress caused by torsion; (**b**) Normal stress caused by axial force and bending moment; (**c**) Normal stress distribution of lower flange of mid-span section.

**Figure 12 materials-15-07920-f012:**
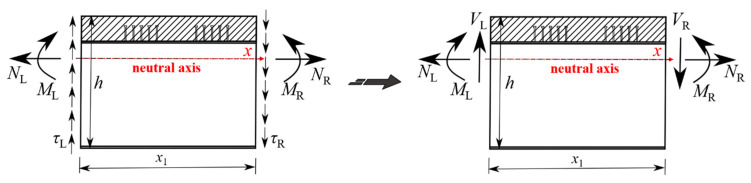
Internal force distribution of steel plate composite girder bridge members.

**Figure 13 materials-15-07920-f013:**
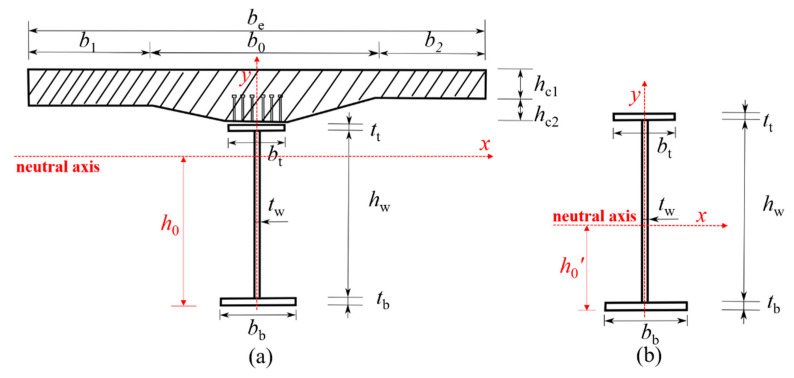
Calculation diagram of section geometric characteristics. (**a**) Composite twin-girder bridge in the positive bending moment area; (**b**) Composite twin-girder bridge in the negative bending moment area and small cross beams.

**Figure 14 materials-15-07920-f014:**
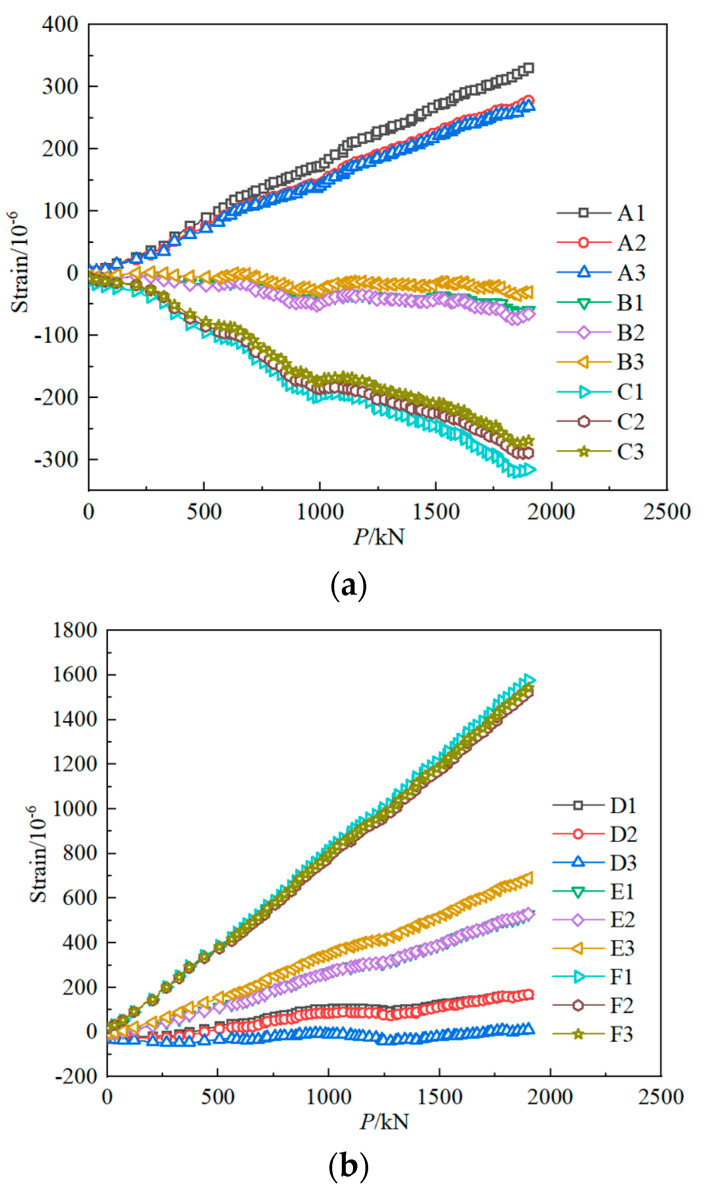
The strain of potential fatigue details in curved composite twin-girder bridge and small cross beam. (**a**) Class I; (**b**) Class II; (**c**) Class III.

**Figure 15 materials-15-07920-f015:**
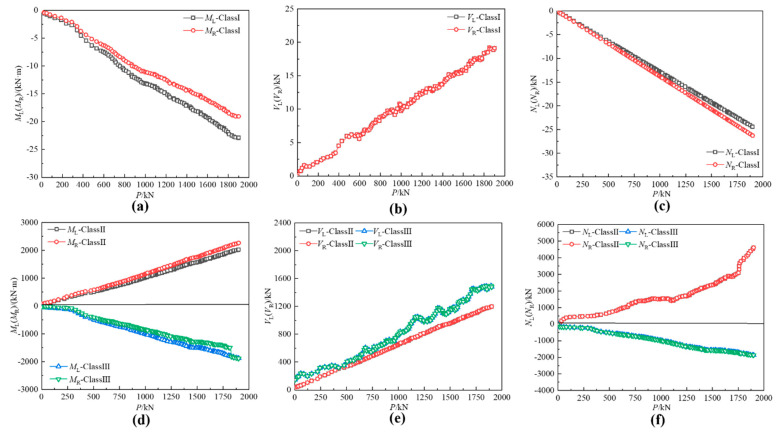
The internal force of potential fatigue details in curved composite twin-girder bridge and small cross beam. (**a**) The bending moment of Class I; (**b**) The shear force of Class I; (**c**) The axial force of Class I; (**d**) The bending moment of classes II and III; (**e**) The shear force of classes II and III; (**f**) The axial force of classes II and III.

**Figure 16 materials-15-07920-f016:**
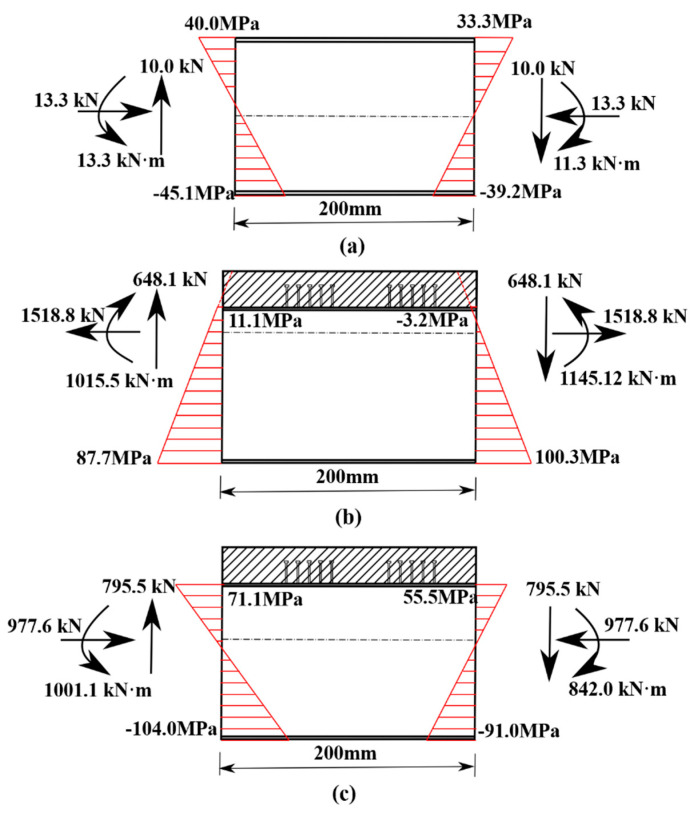
Internal force distribution of classes I–III under *P* = 1000 kN. (**a**) Class I; (**b**) Class II; (**c**) Class III.

**Figure 17 materials-15-07920-f017:**
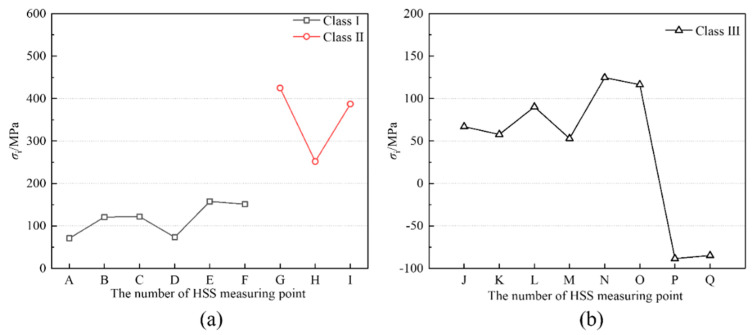
HSS distribution of classes I–III under *P* = 1000 kN. (**a**) Classes I and II; (**b**) Class III.

**Figure 18 materials-15-07920-f018:**
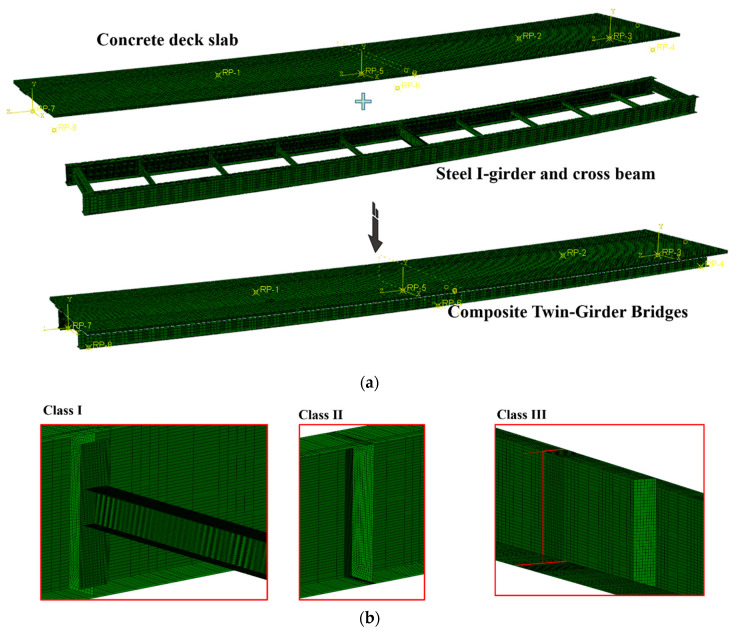
FE modeling of curved composite steel plate twin-girder bridge. (**a**) 3-D global FEM model; (**b**) Modeling of fatigue details

**Figure 19 materials-15-07920-f019:**
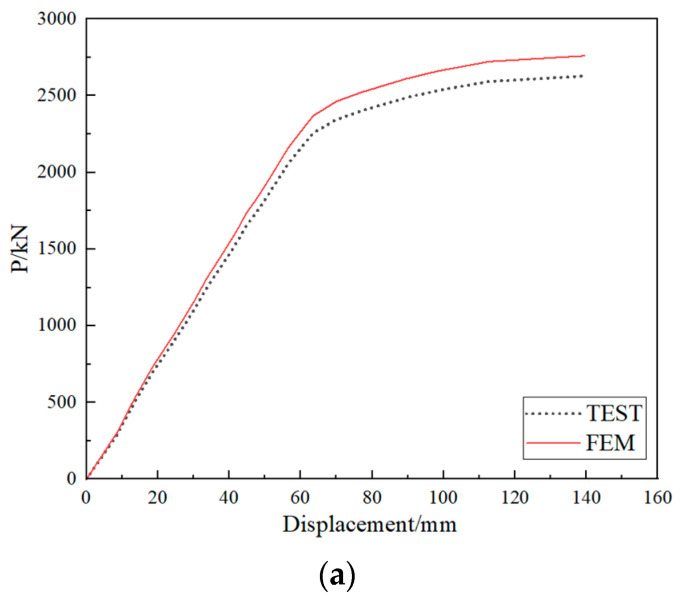
The comparison between the FE and test value. (**a**) Load displacement curves; (**b**) Hot spot stresses curves.

**Table 1 materials-15-07920-t001:** Dimensions of steel I-girders and cross beams (Unit: mm).

Section		Top Flange		Bottom Flange		Web	
		Thickness	Width	Thickness	Width	Thickness	Width
**Steel I-girders**	A	11	400	20	480	10	869
	B	14	400	27	480	10	859
	C	16	400	19	480	12	865
	D	24	400–500	30	480–580	14	846
Small cross beam		10	150	10	150	8	180
Middle cross beam		10	300	10	300	8	480
End cross beam		10	300	10	300	8	480

**Table 2 materials-15-07920-t002:** Materials of steel.

Steel Plate Thickness (mm)	Elastic Modulus *E*_s_ (MPa)	Yield Strength *f*_y_ (MPa)	Ultimate Strength *f*_u_ (MPa)
8	2.06 × 10^5^	474	638
10	2.06 × 10^5^	437	596
12	2.06 × 10^5^	483	642
14	2.06 × 10^5^	422	614
16	2.06 × 10^5^	427	596
20	2.13 × 10^5^	455	589
24	2.02 × 10^5^	412	555
28	2.02 × 10^5^	451	586
30	2.06 × 10^5^	411	548

**Table 3 materials-15-07920-t003:** Materials of concrete.

Concrete Grade	Cube Crushing Strength *f*_cu_ (MPa)	Split Strength *f*_ts_ (MPa)	Elastic Modulus *Ec* (MPa)
C50	52.7	5.02	4.05 × 10^4^
C55	65.6	4.78	3.66 × 10^4^

**Table 4 materials-15-07920-t004:** Loads corresponding to each feature point in the loading process.

No.	Failure Modes	*P*/kN
①	Initial crack appearing in the concrete deck slab	300
②	Full-width crack appearing in the concrete deck slab	1350
③	Yield of the lower flange of the external arc steel I-girder	1811
④	Buckling of the external arc steel I-girder	2603
⑤	Ultimate strength of the whole bridge (peak loading)	2627

**Table 5 materials-15-07920-t005:** Results of strain in classes I–III.

		ε_4 mm_ (10^−6^)	ε_8 mm_ (10^−6^)	ε_12 mm_ (10^−6^)			ε_4 mm_ (10^−6^)	ε_8 mm_ (10^−6^)	ε_12 mm_ (10^−6^)
Class I	A	33.67	32.99	32.69	Class III	J	14.37	4.76	3.62
	B	53.92	51.12	50.74		K	13.11	5.10	3.95
	C	54.64	51.29	50.96		L	20.58	8.07	6.25
	D	35.20	34.41	34.17		M	12.19	4.86	3.76
	E	62.96	55.11	51.30		N	26.06	7.28	4.17
	F	63.12	56.00	55.50		O	27.42	9.67	3.23
Class II	G	253.35	191.55	168.24		P	−36.62	−27.94	−16.86
	H	151.07	114.90	101.00		Q	−29.44	−22.12	−19.22
	I	232.54	177.37	156.64					

**Table 6 materials-15-07920-t006:** Results of HSS in classes I–III.

**Fatigue Details**	**Class I**						**Class II**		
Number of Measuring Points	A	B	C	D	E	F	G	H	I
*σ*_i_/MPa	71.1	121.3	82.2	43.0	157.1	51.2	425.0	251.8	387.2
**Fatigue Details**	**Class III**								
Number of measuring points	J	K	L	M	N	O	P	Q	
*σ*_i_/MPa	67.2	58.3	90.0	53.5	125.1	116.7	−88.1	−85.5	

**Table 7 materials-15-07920-t007:** Category with fatigue details in Eurocode.

Fatigue Details	Nominal Stress Method		Hot Spot Stress Method		Description
	Picture	Detail Category	Picture	Detail Category	
Class I	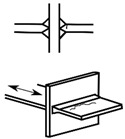	80	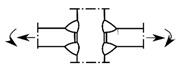	100	Cruciform connections
Class II	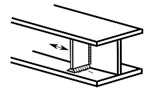	80	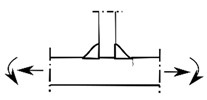	100	Transverse attachments
Class III	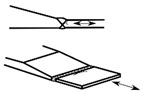	90	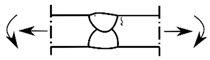	112	Transverse splices

**Table 8 materials-15-07920-t008:** The number of cycles predicted by the NS method and HSS method.

Fatigue Details	Nominal Stress Method		Hot Spot Stress Method		Δ*σ*_h_/Δ*σ*_n_	*N*_n_/*N*_h_
	Δ*σ*_n_	*N* _n_	Δ*σ*_h_	*N* _h_		
Class I	40	34790239	157	516811	3.9	67.3
Class II	170	208426	425	26053	2.5	8.0
Class III	71	4073638	125	1438646	2.4	3.5

**Table 9 materials-15-07920-t009:** Fatigue load model IV in Eurocode.

Order Number	Lorry	Axial Spacing (m)	Equivalent Axial Loads (kN)	Lorry Percentage (%)			Wheel Type
				≥100 km	50 ~ 100 km	<50 km	
①	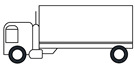	4.5	70 130	20	40	80	A B
②	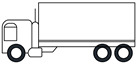	4.2 1.3	70 120 120	5	10	5	A B B
③	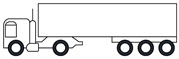	3.2 5.2 1.3 1.3	70 150 90 90 90	50	30	5	A B C C C
④	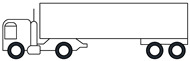	3.4 6.0 1.8	70 140 90 90	15	15	5	A B B B
⑤	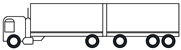	4.8 3.6 4.4 1.3	70 130 90 80 80	10	5	5	A B C C C

**Table 10 materials-15-07920-t010:** Maximum HSS amplitude of potential fatigue details under fatigue load model IV.

	HSS Amplitude *σ*_i_ (MPa)					Remaining Fatigue Life N/Cycle
Potential Fatigue Details	①	②	③	④	⑤	
Class I	32.8	35.3	42.0	38.4	40.7	225910335
Class II	41.3	45.7	68.9	57.1	62.2	32201336
Class III	38.9	39.5	49.1	42.8	46.1	311946213

## Data Availability

The date used to support the findings of this study are included in the paper.

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
