# Peer review of "Experimental Study on Hot Spot Stresses of Curved Composite Twin-Girder Bridges"

_materials, 2022, doi:10.3390/ma15227920_

Round 1
Reviewer 1 Report
In this study, the specimen of 1:2 scale curved composite twin-girder bridge in accordance to the design scheme of Xizhen Bridge in China is tested to determine the nominal stress (NS) and hot spot stress (HSS) at three details, i.e. cruciform connections, transverse attachments and transverse splices. The determined HSS is used for the fatigue life estimation according to Eurocode 3. The fatigue lives estimated by using the NS and HSS are compared. Finally, the fatigue assessment of the target bridge is conducted by using the HSS obtained with FE analysis.
The results of the test using the large-scale specimen is valuable. However, this reviewer cannot find any novelties in the manuscript except the test results. The fatigue assessment process shown in the manuscript is generally used in the design practice according to Eurocode 3. Since the target is the only one bridge, the manuscript seems to be just a case study for the bridge. No general conclusions are obtained. The manuscript is not well-organized and sufficient information is not given for the readers to understand the contents. In addition, there is a critical mistake in the determination of the detail category shown in Table 6. It leads to the different conclusion from the present manuscript. Consequently, this reviewer cannot recommend publishing this article.
Reviewer 2 Report
The paper presents a very interesting large-scale test of a twin-girder 2-span continuous composite bridge. Such a tests are very rarely performed due to very big costs and effort. This is why the paper has a very big potential. Also the topic is very interesting: confirmation of FE methods for evaluation of hot spot stress with large-scale measurements.
However, the presentation of the research is poor and needs to be improved significantly before acceptance for publication. In current form almost all results are presented in such a way, that it is totally impossible for the reader to check and follow the calculations. Below all remarks are listed sequentially, in the order in which they appear in the text:
· Fig. 2 (d) – what rebars were placed in the slab? important over intermediate support, because it influences the distribution of stiffness along the length of the girders. After cracking of concrete the stiffness of slab is governed by reinforcement.
· Line 119 and fig. 3 (class II) – is the stiffener welded by full penetration weld? Usually fillet welds are used. (this is not a critical remark, I just wanted to confirm, it is difficult to distinguish in picture, but the weld looks like fillet one)
· Line 132 – what kind of wet joints are used? It is not presented where are these joints.
· Line 142 – is it a 4 points bending? 2 jacks + 3 supports = 5 points bending per 1 girder. There are 2 girders, so 4 jacks + 6 supports = 10 points bending? It should be rewrite.
· Line 172 – for welded details with high residual stresses cracks are appearing also under compression. But their propagation is not so quick.
· Fig. 7 – it is not presented where are all these points A-J located at a girder length? Only locations in cross-section of girders are presented.
· Line 200 and Fig. 8 – this is strange that cracking of concrete in the M- regions did not affect the P-disp curve. Only 1 crack in concrete slab appeared? What was a crack width / cracks pattern? What was a reinforcement layout? It requires clarification / explanation. Usually after cracking occurs the stiffness decreases and deflection increases. In Fig. 8 I see straight line from 0 up to 2000 kN.
· Line 201 – what buckling? it requires more specific explanation what form of buckling occurred?
· Fig. 8 – is P for 1 jack? Or 2? Or all 4?
· Eq. (1) – this is true only in uniaxial state!
· Eq. (3) – τLdτ? Or rather τLdh where h states for height of the web?
· Eq. (4) – here is an obvious error. ML is in [kNm] and integral(τL) gives shear force which is in [kN]. Summation of values with different units?!
· Lines 264 – 279 (and Eq. (7)-(13)) – this in not needed in the paper. It is enough to describe that cross-sectional parameters are calculated with a modular-ratio method (but this is just my opinion).
· Fig. 12 – it should be shown in which sections of which elements these internal forces are calculated. And also on basis of which measurements? Now it is unclear what exactly is M_L and M_R and so on... what is _L and _R? sketches are necessary! Moreover internal forces presented in Fig. 12 are results of some calculations. Input information were strain measurements. It is needed to provide these measurements. This is the only way for the reader to follow the calculations on one’s own. Also a simple check from solving a static system of the girder would be valuable to confirm obtained forces.
· Line 306 and Fig. 12e – how is it possible to obtain a shear force of 6481 kN? shear force can be easily derived from solution of static system. Forces obtained on basis of SGs measurements and further calculations should be approx. convergent with static system solution. Now it is clearly visible that there are calc. errors. If external force is about 1000 kN, shear force in the girder cannot be 6481 kN…! This proves that all forces calculated in the paper are incorrect.
· Fig. 13 – as above. Where are these sections? in which points of girders?
· Tab. 5 – how these stress are obtained? for each point A-Q strip of SGs were used so 3 strains were measured = 3 stress are obtained (acc. to previous description). Here only 1 stress is shown, as HSS. But how it was calculated? Why stress distribution in 3 measured points were not shown and discussed? Such a way it is not possible to verify correctness of the solution and, in general, the paper does not provide scientific data. The results from SGs strips should be plotted.
· Line 337 – no. Due to high residual stress compressive stress can cause fatigue crack initiation as well.
· Entire chapter 4 is a description of the well known method. It is suggested to shorten or rewrite it.
· Lines 415 - … It should be shown / explained how SCF factors were calculated. This is crucial for further investigations.
· Tab. 6 – for class I in HSS method should be category 100 not 90? See tab. B.1 detail 3 of EN1993-1-9. In the paper this detail is made with full-penetration butt welds, not fillet welds? For class II in NS method should be category 80 not 90? See tab. 8.4 detail 7 of EN1993-1-9?
· Tab. 7 – it should be explained how the HSS are calculated? For class II stress range of 170 MPa is very big. From what kind of fatigue loads? it is not explained.
· Line 433 – there are headed studs. Is the steel and concrete part connected only by means of friction?
· Lines 429 – 433 – how cracking of concrete is enabled in FEM? is tension stiffening effect included?
· Fig. 16b – such a coarse mesh is not relevant for HSS assessment. FE strains in particular regions as well as procedure leading to evaluation of HSS should be shown. It seems that IIW recommendations are not fulfilled... so how HSS were determined? These information are missing, and are critical in this paper. In my opinion the goal should be to compare HSS obtained from tests and FEM. In Fig. 17b I see such a comparison. But it is neither presented how HSS are obtained from test nor from FEM.
· Line 463 – why safe? It depends on the traffic volume... for highways with high flow rates of lorries it is possible to have 2*10^6 lorries per 1 year and per 1 slow lane!
· Tab. 9 – as above. Unknown method of HSS calculation.
· Line 489 – the curvature of this bridge is small, steel girders are connected to concrete slab which has a great torsional stiffness. Actually, implementation of bridge curvature into test setup does not bring any benefits, and can cause only increase of doubts during evaluation of results.
· Line 503 – see remark 26. The statement that the bride is safe is not true.
Round 2
Reviewer 1 Report
The quality of the manuscript is improved to some extent. However, this reviewer still cannot find any novelties in the manuscript except the test results since the contents essentially is the same as the previous manuscript. Consequently, this reviewer’s evaluation result does not change.
Author Response
Please find the response in the attachment.

Reviewer 2 Report
Remarks are included in the attached file.
I also have an opinion that in research papers results of experiments should be presented also as a raw data (or after simple transformations). In this case we have all results provided at once as the final values that it is not possible to check / follow calculations. In my opinion it disqualifies the paper at this moment.

Author Response

(The authors gave the same response as above.)

Round 3
Reviewer 2 Report
Minor revision is recommended acc. to attached file. No re-review is needed afterwards.

Author Response
Thanks for the excellent comments. Please find the response to comments in the attachment.
